# Animal Disease Models and Patient-iPS-Cell-Derived In Vitro Disease Models for Cardiovascular Biology—How Close to Disease?

**DOI:** 10.3390/biology12030468

**Published:** 2023-03-20

**Authors:** Nanako Kawaguchi, Toshio Nakanishi

**Affiliations:** Department of Pediatric Cardiology and Adult Congenital Cardiology, Tokyo Women’s Medical University, Tokyo 162-8666, Japan; pnakanis@gmail.com

**Keywords:** in vivo, in vitro, disease model, iPSC, pulmonary hypertension, humanized animal, machine learning

## Abstract

**Simple Summary:**

Currently, rodents and pigs are the primary disease models used in cardiovascular research. Generally, larger animals that are more closely related to humans are better-suited disease models. However, they can have restricted or limited use because they are difficult to handle and maintain. Animal welfare laws regulate experimental animals. Different species have different mechanisms of disease onset. Organs in each animal species have different characteristics depending on their evolutionary history and living environment. For example, mice have higher heart rates than humans. Nonetheless, preclinical studies have used animals to evaluate the safety and efficacy of human drugs because no other complementary method exists. We need to evaluate the similarities and differences in disease mechanisms between humans and experimental animals, and the translation of animal data to humans can be helpful in this evaluation. In vitro disease models can also be used as human disease models. Three-dimensional human cardiomyocyte cultures are generated from patient-derived induced pluripotent stem cells (iPSCs), which are genetically identical to the derived patient. In this review, we explore the possible use of animal disease models, iPSC-derived in vitro disease models, humanized animals, and the recent challenges of machine learning in making the models more similar to human diseases.

**Abstract:**

Currently, zebrafish, rodents, canines, and pigs are the primary disease models used in cardiovascular research. In general, larger animals have more physiological similarities to humans, making better disease models. However, they can have restricted or limited use because they are difficult to handle and maintain. Moreover, animal welfare laws regulate the use of experimental animals. Different species have different mechanisms of disease onset. Organs in each animal species have different characteristics depending on their evolutionary history and living environment. For example, mice have higher heart rates than humans. Nonetheless, preclinical studies have used animals to evaluate the safety and efficacy of human drugs because no other complementary method exists. Hence, we need to evaluate the similarities and differences in disease mechanisms between humans and experimental animals. The translation of animal data to humans contributes to eliminating the gap between these two. In vitro disease models have been used as another alternative for human disease models since the discovery of induced pluripotent stem cells (iPSCs). Human cardiomyocytes have been generated from patient-derived iPSCs, which are genetically identical to the derived patients. Researchers have attempted to develop in vivo mimicking 3D culture systems. In this review, we explore the possible uses of animal disease models, iPSC-derived in vitro disease models, humanized animals, and the recent challenges of machine learning. The combination of these methods will make disease models more similar to human disease.

## 1. Introduction

Animals are widely used as disease models in medical research, including for the evaluation of the pathophysiology of the disease as well as drug effects and side effects before clinical usage [1]. For this purpose, larger animals, especially those that are evolutionarily close to humans and have a similar size as humans, such as pigs, are generally more suitable than small animals, such as zebrafish, mice, and rats. Humans and certain animals share characteristics due to similarities in various chemicals that compose these organisms. Furthermore, all living organisms are considered to share an ancient origin. However, the evolution of organisms varies because of natural selection, as postulated by Darwin [2]. As a result, differences commonly exist between the organs and tissues of animals, which have adapted to different environments. In the field of cardiovascular research, different environments may result in different heart structures; for example, zebrafish have one atrium and one ventricle, whereas mice and humans have two atria and two ventricles. The characteristics of heart function also vary [3,4,5]. Nevertheless, mice and rodents are most often used as models of cardiovascular disease because they are easy to handle, have lower upkeep costs, and maintain stricter strains, resulting in fewer genetic differences. To establish cardiovascular disease models for diseases, such as myocardial infarction (MI), thoracic surgical procedures, such as ligation of the coronary artery, are performed because the function and structure of the heart are affected by systemic, coronary, and pulmonary circulation [6].

For example, one way in which pulmonary circulation changes heart function is the pulmonary arterial hypertension (PAH) model. We recently established a PAH model in rats in an attempt to identify new drugs for the treatment of this disease. PAH is characterized by endothelial dysfunction, chronic inflammation of the pulmonary artery, smooth muscle cell proliferation, pulmonary arteriolar occlusion, resistance to the apoptosis of pulmonary vascular cells, and pulmonary vascular remodeling. Right ventricular overload, right heart failure, and death may result from these severe pathologies [7,8,9]. However, no effective treatment has been discovered. We established this disease model via a low oxygen (10%) treatment (hypoxia) and monocrotaline (MCT) injections to induce inflammation in pulmonary vascular cells. Other laboratories have used hypoxia only, a combination of hypoxia and MCT or Sugen treatment to establish PAH models [10,11,12,13]. Although we attempted to establish a disease model similar to that of human patients, we were unable to establish the severe phenotype observed in human patients. Although complex occlusion and the recanalization of the pulmonary small artery (plexiform) are observed in human patients with severe PAH [8,14], it is difficult to establish rat models of severe PAH.

Other methodologies used to establish human disease models have recently made significant advances. For example, many in vitro disease models have been established [15,16,17] since induced pluripotent stem cells (iPSCs) were first developed by Takahashi and Yamanaka [18,19]. However, the immaturity of cardiomyocytes derived from iPSCs has been reported [20,21]. This is one of the problems associated with iPSC-derived cardiomyocytes, with symptoms often appearing after birth. The solution to this problem is to mature the cells; for example, a longer culture time may be used to obtain more mature iPSC-derived cardiomyocytes (iPSC-CMs) [22]. The isolation of mature iPSC-CMs, using markers such as SIRPA [23] or the selection of iPSC-CMs with a lactate-containing culture to eliminate other cell types, has been performed [24]. Using these methods and a kit for selection using magnetic beads (Miltenyi), we recently established LQT3-patient-derived iPSCs and differentiated them into cardiomyocytes [25]. This also encouraged us to seek more adequate in vitro disease models.

In this review, we describe animal disease models using zebrafish, mice, and rabbits; compare them with human disease; and discuss how we can make these animal disease models more similar to human patients. We then describe the in vitro disease models. Furthermore, we discuss how we can make these models more useful. The use of animal and in vitro disease models, as well as humanized animals and machine learning, is likely to help in the creation of disease models that mimic human disease more accurately, the combination or integration of which will promote the development of better disease models.

## 2. Animal Disease Models

### 2.1. Zebrafish

Zebrafish are often used in developmental biology because they are transparent, making it easy to observe their organs. Additionally, they have a simpler circulatory system than mammals. Although the structure of the heart of zebrafish appears different, some of their electrocardiogram parameters are more comparable to those of humans than those of mice; heart rates are 120–180 beats per minute (bpm) in zebrafish, 300–600 bpm in mice, and 60–100 bpm in humans [3,4,5]. Both humans and zebrafish generate a repolarizing current during action potential (AP), using a similar rapidly activating delayed rectifier potassium current. Therefore, QT syndrome mutations have been successfully modeled in zebrafish [26,27,28]. Peal et al. [29] used these models to identify the suppressors of long QT syndrome. In the case that the disease being modeled is associated with heart rate and/or potassium current dysfunction, zebrafish are more suitable than mice. Suitable animals should be selected to establish disease models that match the symptoms and conditions of the patients.

Another advantage of using zebrafish as a model organism for cardiovascular research is the usage of morpholino, which can be used as an antisense oligonucleotide to knockdown target genes [5]. Simmerton et al. introduced amine via ribose-to-morpholine transformation [30] instead of coupling the hydroxyls required to form the phosphor–ester-inter-subunit linkage of most nucleic acid analogs, which has high costs. Additionally, morpholino is not catalyzed by enzymes because it is an artificial substance. Therefore, it may be effective for longer compared with natural oligos. Moreover, eGFP knock-in zebrafish can also be a good model for tracking angiogenesis [31,32]. Taken together, research suggests that zebrafish are the most suitable model organism for the study of genes associated with heart structural formation.

### 2.2. Rodents

Rodents, though they are small in size, are more genetically and evolutionarily close to human beings than pigs. Furthermore, transgenic technologies are the most developed in mice, making them the most useful organism for developing disease models. In particular, gene editing using embryonic stem cells (ESCs) has been extensively developed in mice. Moreover, mice have been widely studied genetically, and strains with fewer background differences have been established. Recently, advances in genome-editing technologies have led to the development of powerful tools [33]. Cardiovascular diseases are often associated with metabolic perturbations. In this context, Ld^l−/−^ and ApoE^−/−^ mice have been found to mimic human carotid intima–media thickness better than any other mouse models, including *kl/kl* (*klotho* mutant) mice [34]. *Klotho* mutant mice are transgenic mice that resemble aged humans and have short lifespans [35]. Therefore, to examine the function of their genes, knockout and knock-in methods provide a great advantage. Although rats are also useful, in terms of genetic engineering and strain development, rats are not as well developed as mice. However, because their heart size is larger than that of mice and as they are easier to handle, especially when surgical procedures are involved, they are more useful than mice.

### 2.3. Pigs

Pigs are often used not only in animal disease models but also in human transplantation due to their similar body sizes. Indeed, a recent attempt was made to transplant a pig’s heart into a human patient [36]. Furthermore, the valves of the heart in pigs have been used for xenografts [37]. Gabriel et al. [38] demonstrated heart development in pigs and performed gene editing on SAP130, a histone deacetylase repressor complex previously reported to contribute to hypoplastic left heart syndrome. The authors reported that genome editing was also successful in pigs. 

### 2.4. PAH Model

Because human patient tissue samples are difficult to obtain, PAH animal models have been used to study how the disease progresses histologically, as well as to evaluate the efficiency of different medicines. Furthermore, to evaluate PAH, time-course studies and preclinical pharmacological studies are required, and the observation of the tissue-histology of the patients is not usually possible without performing a lung biopsy. To establish a PAH animal model, we subjected rats to hypoxic conditions and/or injected monocrotaline (MCT) on the first day of the experiment. Abe et al. [39] used Sugen (SU) 5416, semaxinib, which is a vascular endothelial growth factor (VEGF) receptor blocker. According to their report, plexiform-like lesions took a long time to form. Therefore, in addition to Sugen injections, the rats were kept under hypoxia for three weeks and under normoxia for two weeks. Abe et al. demonstrated that, in the case of MCT administration, rats died of cardiac/renal dysfunction before plexiform lesions could form. As shown in Figure 1, in the MCT/hypoxia rats that we established [40,41], occlusive lesions were formed due to the hypertrophy of the medial and endothelial layers; however, the histological changes were less complex compared with those in human sections (Figure 1Left) or to the image displayed by Abe et al. Despite this, the right ventricle systolic pressure (Figure 2) mimicked that of human patients (>25 mmHg), and other hemodynamic data were also similar to those of human patients [40,41]. Therefore, functional MCT/hypoxic PAH models appear to capture the characteristics of this disease in humans. Because MCT is less expensive, it has often been used to establish PAH models. Again, suitable animal disease models should be selected; however, researchers may need to compromise based on their cost constraints. We found that the severity of PAH in rat models depended on the time it took to reach 10% hypoxia (data not shown). A rapid reach resulted in a more severe phenotype. Therefore, in addition, environmental conditions also affect the degree of disease severity. 

### 2.5. Making Animal Disease Models More Similar to Human Diseases

We have shown that a comparison is possible between animal disease models and human diseases. The next step may be to make animal disease models more similar to human diseases. The selection of suitable animals may contribute to the development of suitable disease models. For this reason, we must know the similarities and differences between the animal and human heart and the cells within the heart. Joukar comparatively reviewed experimental animals often used for channel activities, APs, and electrocardiograms (ECGs) [42]. The dissimilarity of the AP profile of rodents and mice with that of humans comes from diversity in distribution and the density of potassium channels. The APs and ECGs of human hearts were most similar to those of rabbits; then to guinea pigs; and, thirdly, to rats/mice. However, small animals, such as rats and mice, are often used because of the convenience of handling and lower costs, and as they have fewer ethical problems. Mice in particular are used for gennome editing. Knock-in mutated genes can produce a disease model for diseases owing to mutations. For example, SCN5A, which encodes a sodium channel, has been well characterized. *Scn5a*^G1746R^ knock-in mice recapitulate Brugada syndrome [43]. Homozygotes in these knock-in mice could not be generated. It was previously found that a multicopy suppressor of Gsp1 (MOG-1) binds with Nav1.5 using two-hybrid screening [44]. Yu et al. showed that MOG-1 gene therapy was effective in a Brugada syndrome mouse model [43].

Morotti et al. [45] presented the interesting idea of “cross-species translators”. They analyzed the APs of mice, rabbits, and humans to build suitable translators to quantify electrophysiological responses in ventricular myocytes across species. Next, they fit the electrophysiological responses of mice to those of humans using the translators in order to predict the responses of humans to certain drugs by measuring the response of the mice. They first established AP models in mice, followed by rabbits. Then, they constructed predictors for humans from those of the mice and rabbit models using a computational approach. Furthermore, they validated their translators against experimental data and demonstrated suitability in predicting human responses to drugs from animal models [45]. Interestingly, higher accuracy was observed in rabbit-to-human translation than in mouse-to-human translation. This notion of “translation” was proposed by Gong and Sobie [46], who used a combination of mechanistic mathematical modeling and statistical analysis. Since variations were observed in iPSC-CM, including differences in maturity, and the subtype population of cardiomyocytes was observed even in the same cultures, they proposed a cross-cell-type regression model that predicted adult cardiac myocyte drug responses from iPSC-CM behavior. Therefore, the development of translators across species may represent a solution to these problems.

## 3. In Vitro Disease Models

### 3.1. Engineered Heart Tissues and Organoids

In vitro disease models have been developed since the establishment of iPSCs. Simultaneously, regenerative medicine has been developed using stem cells, such as embryonic stem cells and bone marrow stem cells. Recently, progress has been made in the development of three-dimensional (3D) cell culture techniques. The cells were classically cultured in 2D. Three-dimensional culturing has received increasing attention owing to its ability to recreate in vivo environments more efficiently. Therefore, various engineered heart tissues with 3D structures have been developed. 

As described previously, iPSC-CMs are immature. However, the majority of heart diseases appear after birth and with age. Therefore, mature iPSC-CMs are required for in vitro disease models more than immature iPSC-CMs before birth. To mature the cells, longer culture periods and the addition of various cytokines and hormones are required [47,48], as well as electrical stimulation [49], co-culture with non-CMs [50], and extracellular matrix treatment [51]. Three-dimensional culturing can also be used to obtain mature iPSC-CMs since it is more similar to the in vivo conditions and can be developed with and without scaffolds. Engineered heart tissues (EHT) can be prepared using the scaffolds. Spheroids made by culturing in suspensions or low attachment plates and cell sheets made by culturing on thermo-responsive polymers have also been applied to 3D cultures. Using the EHT system and iPSC-CMs, models for chronic non-hereditary cardiomyopathy, such as heart failure, have been established via chronic norepinephrine stimulation [20]. 

Organoids are 3D organ-like structures that display the characteristics of their corresponding organs and have been used in various disease models [52,53]. According to Tang et al. [52], organoids were first reported in oncology in 1946 as a synonym for teratomas. In the 1960s, organoids were used by developmental biologists to describe organogenesis via the reaggregation of dissociated cells derived from the same organs [52]. Organoids can be composed of primary cells, iPSCs, or ESCs. They can be used as disease models for the development of various medical therapies and studies, including drug discovery. 

Decellularized hearts obtained by dissociating cells within the heart of rats via SDS treatment, which maintains the shape of the heart, is another strategy by which 3D hearts can be obtained, particularly in mice and rats [54]. Decellularized hearts contract after plating neonatal cardiomyocytes. Human hearts may be too large for this method, which seeks to compose a contracting heart. Kitahara et al. recellularized a decellularized heart with mesenchymal stem cells, transplanted it, and observed coronary artery blood flowing using pigs [55]. Furthermore, contracting heart tissues composed via 3D printing were recently developed successfully [56]. 

### 3.2. Multi-Lineage Differentiation of iPSCs

To date, one specific lineage differentiation has been attempted in cardiovascular research, mainly because undifferentiated cells should be removed when differentiated cells are used for regenerative medicine. Cardiomyocytes are key cells in cardiac regenerative medicine to restore heart function. Studies to obtain mature cardiomyocytes have also been conducted. The typical protocols for cardiomyocyte differentiation involve two steps: the induction of mesoderm differentiation and the induction of cardiomyocyte differentiation [57,58,59,60,61]. Therefore, different dermal organs, such as the heart (from the mesoderm) and lungs (from the endoderm), are considered to be induced separately and not simultaneously. However, the heart is connected to the aorta, pulmonary arteries, and veins, and during embryogenesis, these two organs (the heart and lung) can connect and influence each other. Moreover, diseases such as PAH strongly influence lung function as well as right ventricular function. Ng et al. [62] established cardiopulmonary tissues using simultaneous multi-lineage differentiation. This study aimed to understand how the heart and lungs influence each other during embryogenesis. During mouse embryonic development, WNT derived from the second heart field induces the specification of the pulmonary endoderm that, in turn, secretes sonic hedgehog (SHH), which signals back to the heart and regulates proper arterial septation. Therefore, a system that can trace cardiopulmonary co-development is required. For this purpose, cardio-pulmonary micro-tissues (μTs) were established using iPSCs. WNT signals are necessary for cardiac and lung development. However, WNT should be removed to induce cardiomyocyte contraction. They used different concentrations of a WNT agonist for different time periods. Notably, in certain treatments, the μTs made by suspension cultures, cardio-μTs, and pulmonary μTs are produced spontaneously in the same culture dish and later segregated without the addition of other signal molecules, suggesting that WNT can induce endogenous cytokines/growth factors necessary for next-stage development. Endogenous nodal and BMP signals within the culture may have contributed to these differences. This simultaneous multi-lineage differentiation may be similar to in vivo embryonic development. This kind of multi-differentiation may also contribute to a disease model in which multiple organs are involved in the disease.

## 4. Humanized Animals

Humanized animals express human genes and have human cells and tissues. Previously, human genes have been inserted into mice using a gene-targeting knock-in method. However, to transplant human cells or tissues, immune-deficient mice are needed to avoid immunological rejection. Thus, small animals, such as mice, tend to be used. Although human organs are too large to be generated or transplanted into small animals, interspecies organogenesis enables the generation of human organs. Yamaguchi et al. generated a mouse pancreas in rats [63]. To this end, they first established Pdx-1-deficient rat blastocytes, injected mouse PSCs into Pdx-1-deficient rat blastocytes, and created a mouse pancreas in rats through a process known as blastocyst complementation. The pancreas was transplanted into mice with streptozotocin-induced diabetes. As a result, the transplanted organs normalized the blood glucose levels in the host. These results suggest that interspecies organogenesis may contribute to therapeutic regenerative medicine. Taking into account the size of human organs, pigs are suitable for blastocyst complementation. In the cardiovascular field, pigs have been used for xenotransplantation. In a previous study, a pancreas was established using Pdx-1-knockout pigs [64]. However, pig-to-human infections are a concern. According to a review article by Kano et al. [65], the knockout of a major xenograft antigen galactose, a1,3-galactose, in pigs was approved by the U.S. Food and Drug Administration (FDA) as a source for human therapy, including xenotransplantation, which reduced concerns regarding immunological rejection. At present, low chimerism is the most important challenge. Blastocyst complementation can be used not only for human therapy but also in vivo disease models, or at least partially if patient-derived iPSCs can be used to make human organs in the experimental animal. 

### Humanized Mice

As the methodologies for the creation of transgenic mice progress, transgenic mice containing human mutated genes and exhibiting phenotypes similar to those of humans can be readily prepared for use as disease models. Furthermore, owing to recent advances in the development of the gene-editing technology CRISPR/Cas9, mutated knock-in mice are becoming widely used. Recently, Stevens et al. [66] established mouse arrhythmogenic cardiomyopathy (ACM) models by knocking in mutant desmoplakin (*Dsp^R451G/+^*). Desmoplakin is a desmosome structural protein located in intercalated discs. Mutations in this gene are associated with ACM. *Dsp^R451G/R451G^* mice were found to die before E20, highlighting the important role of this protein. The low expression of the Dkp protein in *Dsp^R451G/+^* mice resulted in reduced cardiac performance, increased chamber dilation, and accelerated progression to heart failure, as observed in human patients. Human patients typically have heterogeneous mutations. Therefore, these models are considered to be more similar to humans, since the phenotype appeared in heterogeneous mutations. Notably, Ng et al. [67] not only observed a correlation between ACM and genetic inheritance but also found that *Dsp^R451G/+^* patients had a lower protein level of desmoplakin and connexin 43 than donors without heart disease. However, the protein expression levels of plakoglobin, another desmosome protein, were similar [67]. Ng et al. created EHT by using patient-derived iPSC-CMs. In their study, no significant electrophysiological differences were observed, except for the time to peak of the AP. However, low levels of desmoplakin and connexin 43 were found. The low levels of desmoplakin expression were not due to the stability of the protein but rather due to degradation caused by calpain. Indeed, other mutations that had more interaction sites for calpain showed increased levels of desmoplakin degradation. Taken together, the study of both animal and in vitro disease models may provide more compelling results. 

In contrast, advances made in the production of transgenic mice via gene-editing, human cell or tissue generation, or transplantation in mice have experienced more hurdles, mainly as a result of immunological rejection. In this context, immune-deficient mice, such as NOG mice, should be used. The xenografts of human cells/tissues in NOG mice are mainly used for cancer research. Humanized livers were generated in newly established Nog-TK mice by transplanting human liver cells [68].

Biologically, the same species should be used as the host animal. There may be differences in the transplanted cells/tissues between humanized and regular mice. Because the heart rate differs among animals, a comparison such as molecules and the constitution of the organ/tissues/cells between humanized animals and humans is necessary to obtain information, and some kind of “translation” from animals to patients may help utilize the information obtained from humanized animals. 

## 5. Computer Models

### 5.1. Machine Learning (ML)

Artificial intelligence (AI) has become increasingly familiar in recent years, with popular uses in devices, including Alexa or Siri. Machine learning (ML) is a sub-discipline of AI that involves the processing of input, modeling, and subsequent output. Models are trained using large amounts of data, after which, the best models are selected before testing. Machine learning includes supervised ML and unsupervised ML. Supervised ML involves labeled data that are evaluated by experienced experts. In contrast, unsupervised ML finds similarities and performs clustering independently. In cardiovascular research, since both structural information obtained from images of the heart and electrophysiological data are important for diagnosis and treatment, AI is expected to contribute to this field [69,70,71,72,73,74,75,76,77,78,79]. The algorithms used for ML to select a fitted model and validation methods in electrophysiology were summarized excellently by Trayanova et al. [69]. ML can be a useful tool for disease diagnosis. Indeed, the number of studies using ML in the field has increased rapidly over this past decade. Glass et al. [80] reviewed recent advances in the use of ML for cardiovascular pathology, mainly discussing structural abnormalities that can be associated with pathologies, such as fibrosis and inflammation. This technology has also been developed to find micro-calcifications in mammographies [80,81], which involves simpler images. They proposed taking advantage of computers to analyze clusters of digitalized samples of tissue sections. ML has also been used to estimate ablation sites for the treatment of atrial fibrillation (AF), which is a life-threatening disease. 

This clustering method can also be applied to ECGs. Numerous studies have evaluated electrophysiology data in an attempt to identify arrhythmia for disease diagnosis and treatment using ML [69,70,71,72,73,74,75,76,77,78,79]. In this context, smartwatches and smartphones contribute a large amount of data required for ML. However, the majority of these devices lack sufficient disease data, in contrast to their collection of large amounts of health data. Moreover, data obtained from clinics for these studies do not exhibit sufficient variability, including with respect to race, sex, and age. Furthermore, due to efforts to protect the privacy of patients, data can often only be obtained from a limited number of centers because of limitations in the collection of data from multiple hospitals. The reliability for clinicians is relatively low due to the black-box nature of ML models.

### 5.2. Deep Learning

Deep learning is included in ML. It uses convolutional neural networks similar to human neural networks. The network is composed of layers that are constituted by computer neurons. Interestingly, it has been reported that an ECGage that is >8 years greater than the chronological age is associated with a higher rate of mortality [82]. In 2000, an artificial neural network was used to detect activations during ventricular fibrillation, which resulted in a classification correctness of 92% in the test examples [83], with improvements having been made since [84,85,86]. 

### 5.3. Experimental Data Validation for ML

AI also has the potential to contribute to establishing disease models as an alternative to animal and in vitro disease models. In ML studies, “overfitting” is a common problem that is caused by trying to fit sparse data into a model. In fact, the use of sparse data is a common problem in many models. Earlier, we described how iPSC-CMs are also variable, making it difficult to determine standard iPSC-CMs. Indeed, even normal iPSCs are difficult to determine. To address this, Orita et al. developed a method to distinguish normal iPSCs from abnormal iPSCs using deep learning [87].This system used criteria used by experts who are able to distinguish normal iPSCs from abnormal iPSCs, establishing a method for the automated quality control of iPSCs. 

Furthermore, it is also difficult to evaluate mutated iPSC-CMs. Therefore, efforts have been made to establish standardized iPSC-CMs using ML and compare these with arrhythmic cardiomyocytes using real iPSC-CMs [88] and in silico CMs [89]. Not only were the cells used evaluated, but the methodologies were also compared [76]. The combination of image data and multi-electrode data in ML may improve AF driver detection [77].

Taken together, the combination of machine learning, animal disease models, and in vitro disease models will lead to the development of more robust disease models and, eventually, a reduction in the use of animals for disease modeling. Furthermore, with more training, ML could be used to create disease models that are more similar to human diseases. However, this requires further studies. 

## 6. Summary

Currently, zebrafish, rodents, canines, and pigs are primarily used as disease models in cardiovascular research. However, different species have different mechanisms of disease onset, creating a need for the selection of suitable animals depending on the disease being studied. In this context, the similarities and differences in disease mechanisms between humans and experimental animals also need to be evaluated. The translation from animal to human phenotypes will contribute to the use of disease models. In vitro disease models have been used to study human disease models since the discovery of iPSCs. Cardiomyocytes have been generated from patient-derived iPSCs, which are genetically identical to the derived patients. Researchers have also attempted to develop in vivo mimicking 3D culture systems, such as engineered human tissues (EHTs), spheroids, and organoids. Humanized animals have also been used as animal disease models in disease studies. However, in this context, the differences between humans and animals must be considered. Therefore, comparative studies of donor and host cells and tissues are needed to obtain further information. Finally, in recent years, the use of AI has provided a basis for the creation of artificial disease models that more accurately mimic disease in humans.

## 7. Conclusions

The successful use of animal models for the study of human disease requires the selection of suitable species. In this context, the translation of animal models to human diseases is helpful. In cases where stem cells are used for the creation of in vitro disease models, 3D culturing and the combination of microtissues both contribute to the development of a suitable disease model. Multi-lineage differentiation may have the advantage of using pluripotent stem cells. Furthermore, AI continues to make important contributions to this field. 

## Figures and Tables

**Figure 1 biology-12-00468-f001:**
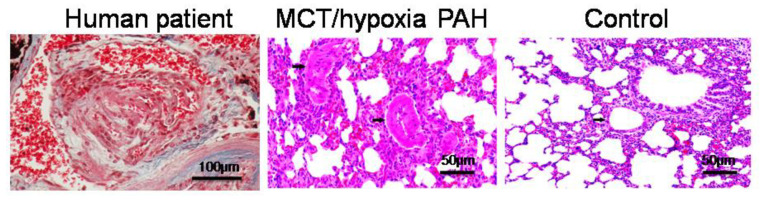
Histological section of the pulmonary artery. (**Left**) Human patient. The section was stained using the Masson stain method. Scale bar, 100 µm. (**Middle**) Rat PAH model and (**Right**) control. The sections were stained with hematoxylin and eosin. Scale bar, 50 µm.

**Figure 2 biology-12-00468-f002:**
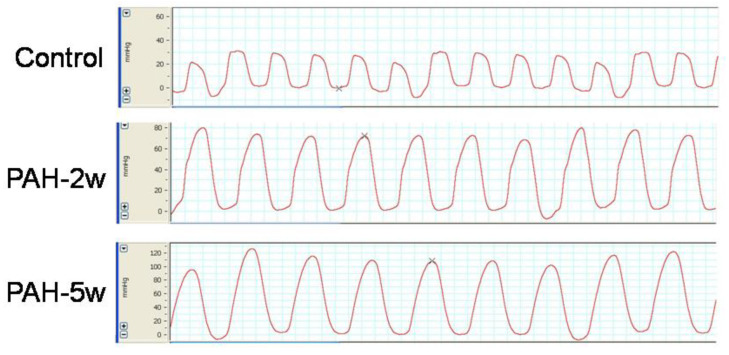
Right ventricle systolic pressure in rat PAH models.

## Data Availability

The data presented in this article are available from *Experimental and Therapeutic Medicine* (Zhang, T.; Kawaguchi, N.; Hayama, E.; Furutani, Y.; Nakanishi, T. High expression of CXCR4 and stem cell markers in a monocrotaline and chronic hypoxia-induced rat model of pulmonary arterial hypertension. *Exp. Ther. Med.*
**2018**, *15*, 4615–4622. https://doi.org/10.3892/etm.2018.6027. PMID: 29805477; PMCID: PMC5952071 [40].) and *Respiratory Research* (Zhang, T.; Kawaguchi, N.; Yoshihara, K.; Hayama, E.; Furutani, Y.; Kawaguchi, K.; Tanaka, T.; Nakanishi, T. Silibinin efficacy in a rat model of pulmonary arterial hypertension using monocrotaline and chronic hypoxia. *Respir. Res.*
**2019**, *20*, 79. https://doi.org/10.1186/s12931-019-1041-y. PMID: 31023308; PMCID: PMC6485095 [41]).

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
