# Peer review of "Animal Disease Models and Patient-iPS-Cell-Derived In Vitro Disease Models for Cardiovascular Biology—How Close to Disease?"

_biology, 2023, doi:10.3390/biology12030468_

Round 1

Reviewer 1 Report

Comments

This is a very interesting paper and it was a pleasure to read it. I have added a few comments below that may be useful to further improve this work. Thank you for all the hard work you have put so far.

·        Long portions of the introduction do not have references, which is unusual in a review piece. For example, the first paragraph only cites one source, but contains facts about heart rates and interspecies differences – could the authors provide more reliable references to support these statements?

·        The sentence starting on line 62-65 is correct, but reads as being out of place as it follows from action potential shapes. It is unclear if this is presented here as it refers to a specific animal model. Should subheadings be used to divide electrophysiological differences from anatomical?

·        Sentence from line 76-80 needs to be revisited and proofread, as it is not clear what the authors are stating.

·        Lines 120 and 131 are missing the year of publication – cannot check with reference list due to it being in a different format/referencing style.

·        The figure legends are not descriptive of what is being shown. These should be expanded further to be understood. Figure legends are also missing the scale size. Were these previously published? If so, the source should be presented in the legend too.

·        Figure 2 – control is misspelt

·        It is unclear what the numbers on lines 174 and 175 refer to – are these animal models or citations?

·        Line 181-186 – authors should expand on the human results obtained from regressing from iSPC-CM models. The latter are known to be immature, so how many parameters are these computational models taking into account to predict the behaviour? What is the accuracy of this model? Have these been validated experimentally?

·        Line 197 needs to be reworded in order to have a clearer meaning.

·    Culture with non-CMs – many studies have been conducted on this using different cell types. It is highly recommended the authors show examples of different cells used and why those were selected.

·        Portions of text are not supported by literature.

This is a very interesting review that compiles information on a lot of topics related to PAH. In places, some introduction is given to the topic, but it lacks depth. It is not clear from the review whether the authors are investigating the differences among different animal models based on genetics, electrophysiology or other. The authors mention PAH and previous work done on this, but the review does not have it as a focus. Some portions of the review are not clear as they mix different maturation markers in iPSC.

I believe this work is of high quality, it just needs to be more organised – perhaps with different headings.  

Author Response

Thank you very much for the constructive comments. The revised manuscript has improved a lot owing to your comments. The reviewer comments are included below in bold red. Our replies are shown in black.

This is a very interesting paper and it was a pleasure to read it. I have added a few comments below that may be useful to further improve this work. Thank you for all the hard work you have put so far.

Response: Thank you for your encouraging comments. We appreciate the time taken to review our manuscript. We agree that the comments have greatly improved our manuscript.

  • Long portions of the introduction do not have references, which is unusual in a review piece. For example, the first paragraph only cites one source, but contains facts about heart rates and interspecies differences – could the authors provide more reliable references to support these statements?

Response: We have added more references to the Introduction, as well as throughout the revised manuscript.

  • The sentence starting on line 62-65 is correct, but reads as being out of place as it follows from action potential shapes. It is unclear if this is presented here as it refers to a specific animal model. Should subheadings be used to divide electrophysiological differences from anatomical?

     Response: We have deleted these sentences because a similar description is provided later in the manuscript.

      Sentence from line 76-80 needs to be revisited and proofread, as it is not clear what the authors are stating.

Response: We have revised these sentences in the revised manuscript as “Right ventricular overload, right heart failure, and death may result from these severe pathologies [7–9]. However, no effective treatment has been discovered. We established this disease model via low oxygen (10%) treatment (hypoxia) and monocrotaline (MCT) injection to induce inflammation in pulmonary vascular cells. (Lines 64-67 in the revised version)

  • Lines 120 and 131 are missing the year of publication – cannot check with reference list due to it being in a different format/referencing style.

Response: We apologize for this mistake. We have deleted these lines from the revised manuscript.

  • The figure legends are not descriptive of what is being shown. These should be expanded further to be understood. Figure legends are also missing the scale size. Were these previously published? If so, the source should be presented in the legend too.

Response: We have corrected our manuscript and provided more information as in the below. We have included a scale bar that indicates the size. To clarify, the photograph shows a section from the PAH patient in our university. Figure 1. Histological section of the pulmonary artery. (A) Human patient. The section was stained using the Masson stain method.Scale bar, 100 µm  (B) Rat PAH model and (C) control. The sections were stained with hematoxylin and eosin. Scale bar, 50 µm.

  • Figure 2 – control is misspelt

Response: We have corrected the spelling in the revised manuscript.

  • It is unclear what the numbers on lines 174 and 175 refer to – are these animal models or citations?

Response: These numbers are referring to proteins.

  • Line 181-186 – authors should expand on the human results obtained from regressing from iSPC-CM models. The latter are known to be immature, so how many parameters are these computational models taking into account to predict the behaviour? What is the accuracy of this model? Have these been validated experimentally?

Response: We have revised this section for language and improved clarity (Lines 175-189 in the revised version). These parts are about AP of ventricle myocoytes across species, not of iPSC-CM. However, their translation notion and methods came from the work of Gong and Sobie.

  • Line 197 needs to be reworded in order to have a clearer meaning.

Response: Line 197 in the original version is “In vitro disease model”. We focus on in vitro disease models in this section..

  •  Culture with non-CMs – many studies have been conducted on this using different cell types. It is highly recommended the authors show examples of different cells used and why those were selected.

Response: Because real heart cardiomyocytes may come into contact with non-cardiomyocytes or the extracellular matrix, iPSC-CMs may need these cells or the extracellular matrix. We have revised the text accordingly.

  • Portions of text are not supported by literature.

Response: We have included further references to our revised manuscript.

This is a very interesting review that compiles information on a lot of topics related to PAH. In places, some introduction is given to the topic, but it lacks depth. It is not clear from the review whether the authors are investigating the differences among different animal models based on genetics, electrophysiology or other. The authors mention PAH and previous work done on this, but the review does not have it as a focus. Some portions of the review are not clear as they mix different maturation markers in iPSC.

Response: We have made improvements to our text accordingly. The comments are excellent. However, it may be our next work to try to make the disease models closer to the real disease in patients. We thought we had to evaluate the present disease models and had to consider how we could do that. That was our motivation to write this manuscript.

I believe this work is of high quality, it just needs to be more organised – perhaps with different headings.  

Response: We have organized the text better in the revised manuscript. We appreciate your kind and encouraging comments.

Reviewer 2 Report

The authors provided a very short review on PAH or cardiovascular animal models. The contents are organized in a poor format with many typos and grammatic problems. How animal models are used in the field of cardiovascular biology and medicine is poorly presented. I could not learn from the review on how to establish or characterize animal models, expect that the rat model the authors generated. I think the authors should include more contents on the zebrafish, rodents, rabbits or pigs in detail, either genetically or surgically. Additionally, a comparison of these models should also present to tell the readers which animal model is suitable to study any specific topic in the field. The machine learning section is very irruptive without proper stating the background and other diagnostic approaches.

Minor:

1. Zebrafish not "zebra fish".

2. mouse is also one of the rodents.

3. Rodents, though they are small in size, are more genetically and evolutionarily close to human beings than pigs does. 

4. What is mechanical learning? I know it's one of the machine learning method. But the content is presented only with the headline 'mechanical learning', then machine learning. Should the authors clarify this?

Author Response

Thank you for the constructive comments. The revised manuscript has improved a lot owing to your comments. The reviewer comments are included below in bold red. Our replies are in black.

The authors provided a very short review on PAH or cardiovascular animal models. The contents are organized in a poor format with many typos and grammatic problems. How animal models are used in the field of cardiovascular biology and medicine is poorly presented. I could not learn from the review on how to establish or characterize animal models, expect that the rat model the authors generated. I think the authors should include more contents on the zebrafish, rodents, rabbits or pigs in detail, either genetically or surgically. Additionally, a comparison of these models should also present to tell the readers which animal model is suitable to study any specific topic in the field. The machine learning section is very irruptive without proper stating the background and other diagnostic approaches.

Thank you for your comments. We have made improvements to the organization of our manuscript in the revised version.

Minor:

  1. Zebrafish not "zebra fish".

Response: This has been corrected in the revised version.

  1. Mouse is also one of the rodents.

Response: This is a very important correction.

  1. Rodents, though they are small in size, are more genetically and evolutionarily close to human beings than pigs does. 

Response: This is a very important correction.

  1. What is mechanical learning? I know it's one of the machine learning method. But the content is presented only with the headline 'mechanical learning', then machine learning. Should the authors clarify this?

Response: We have changed “mechanical learning” to “machine learning” in the revised manuscript.

Round 2

Reviewer 2 Report

The authors have revised their manuscript extensively. I do not have any comments left.